# Clinical Outcomes Associated with Monotherapy and Combination Therapy of Immune Checkpoint Inhibitors as First-Line Treatment for Advanced Hepatocellular Carcinoma in Real-World Practice: A Systematic Literature Review and Meta-Analysis

**DOI:** 10.3390/cancers15010260

**Published:** 2022-12-30

**Authors:** Huimin Zou, Qing Lei, Xin Yan, Yunfeng Lai, Carolina Oi Lam Ung, Hao Hu

**Affiliations:** 1State Key Laboratory of Quality Research in Chinese Medicine, Institute of Chinese Medical Sciences, University of Macau, Macao, China; 2School of Public Health and Management, Guangzhou University of Chinese Medicine, Guangzhou 510006, China; 3Department of Public Health and Medicinal Administration, Faculty of Health Sciences, University of Macau, Macao, China

**Keywords:** advanced hepatocellular carcinoma, first-line, immune checkpoint inhibitors, clinical outcomes, meta-analysis

## Abstract

**Simple Summary:**

Hepatocellular carcinoma (HCC) is one of the most malignant tumors. Immune checkpoint inhibitors (ICIs)-based therapy has recently been demonstrated to greatly ameliorate survival outcomes in advanced HCC. The objective of this research was to evaluate clinical outcomes of ICIs-based monotherapy and combination therapy as first-line treatment of adults with advanced HCC in real-world practice by conducting a systematic literature review and meta-analysis. These findings could supplement the evidence for treatment strategies of advanced HCC and inform study design in future real-world studies of HCC.

**Abstract:**

Background: Immune checkpoint inhibitors (ICIs)-based therapy has recently been demonstrated to greatly ameliorate survival outcomes in advanced hepatocellular carcinoma (HCC). We aimed to evaluate clinical outcomes of ICIs-based monotherapy and combination therapy as first-line treatment of adults with advanced HCC in real-world practice by conducting a systematic literature review and meta-analysis. Methods: PubMed, Web of Science, and Embase were searched up to 25 April 2022. Retrospective or prospective real-world studies evaluating progression-free survival (PFS), overall survival (OS), objective response rate (ORR), disease control rate (DCR), and treatment-related adverse events (TRAEs) of patients with advanced HCC receiving first-line ICIs-based therapy were included. Results: Of 7805 studies retrieved, 38 were deemed eligible for inclusion. For patients receiving first-line ICIs-based therapy in real-world practice, the pooled median PFS and OS were 7.03 (95% CI: 5.55–8.51) and 14.39 (95% CI: 10.91–17.86) months. The ORR and DCR were 0.432 (95% CI: 0.327–0.538) and 0.756 (95% CI: 0.677–0.836), according to mRECIST 1.1, 0.317 (95% CI: 0.218–0.416) and 0.740 (95% CI: 0.644–0.835), judged by RECIST 1.1. The best outcomes of survival and response rate were observed in ICIs-based combination therapy of ICIs, TKIs, plus LRTs. Furthermore, ORR, DCR judged by mRECIST 1.1, and PFS could be potential prognostic factors for OS. Conclusions: This research revealed diversified first-line ICIs-based therapies for advanced HCC in real-world practice. Future studies are needed to adopt prospective, multicentric and comparative designs to test the ICIs-based combination therapies, especially triple therapies of ICIs, TKIs, plus LRTs.

## 1. Introduction

Liver cancer has become the second most frequent cause of cancer-related deaths globally, only after lung cancer [1]. Hepatocellular carcinoma (HCC) is the most common pathological type of primary liver cancer, accounting for 75−85% of cases [2]. Asian patients suffer from the highest incidence of HCC, and Chinese patients, in particular, account for more than half of the new cases annually and bear great impact on their economic burden and quality of life [3,4]. The underlying risk factors of HCC include hepatitis B virus (HBV) and hepatitis C virus (HCV) infection, cirrhosis, smoking, alcohol consumption, non-alcoholic fatty liver disease (NAFLD), obesity, type 2 diabetes, and exposure of aflatoxin [5,6,7,8]. These risk factors affect patients and eventually result in the occurrence of HCC, owing to the sustained inflammation in the liver [5,9,10].

For HCC at an early stage, surgical treatment; locoregional treatment (LRT), mainly consisting of stereotactic body radiation therapy; radiofrequency ablation; and trans-arterial chemoembolization (TACE) can significantly improve the progression-free survival (PFS) and overall survival (OS) of patients [11]. However, most cases were diagnosed as advanced stages owing to the existed vascular invasion or extrahepatic metastases, and thus the majority of them could not benefit from the above interventions, including surgery and other LRTs. Additionally, the clinical outcome of classic systemic therapy was usually not satisfactory. For instance, chemotherapy was usually associated with adverse events and poor prognosis, and the efficacy of molecular targeted therapy has been reportedly limited [12,13,14]. Therefore, it is urgent to develop novel strategies for the treatment of advanced HCC.

In the last decade, monotherapy and combination therapy of immune checkpoint inhibitors (ICIs) have greatly improved the survival outcomes in some solid and hematological tumors [15,16,17]. It is worth noting that several ICIs-based therapies, such as atezolizumab plus bevacizumab, pembrolizumab, and nivolumab, have been approved as first-line or subsequent-line systemic therapy for advanced HCC based on the clinical trials, such as IMbrave150, KEYNOTE-224, and CheckMate-040 [18,19,20].

However, the comprehensive review of clinical outcomes in real-world practice of ICIs-based therapy for advanced HCC remains underreported. Unlike clinical trials, more diversified patient samples are included in real-world practice, which not only supports data from clinical trials but may also supplement the current understanding about the clinical outcomes. The evidence from real-world practice can also provide reference to inform healthcare policy and disease management [21]. Two previous studies evaluated the efficacy of ICIs in HCC by analyzing the data derived from clinical trials and cohort studies published before 2020; however, they did not focus on a specifically systematic analysis of first-line therapy for advanced HCC in real-world practice [22,23].

Thus, the objective of this study was to evaluate clinical outcomes associated with ICIs-based monotherapy and combination therapy as first-line treatment for adults with advanced HCC in real-world practice by conducting a systematic literature review and meta-analysis. It is expected that the findings could supplement the evidence for treatment strategies of advanced HCC and inform study design in future real-world studies of HCC.

## 2. Methods

This systematic literature review and meta-analysis complied with the Preferred Reporting Items for Systematic Reviews and Meta-Analyses (PRISMA) statement [24,25]. The research protocol had been registered with PROSPERO (CRD42021288188).

### 2.1. Eligibility Criteria

The study selection criteria, according to the PICOS approach, were presented in Appendix A [26]. Real-world utilization published in English were included regardless of whether they were prospective or retrospective design. The patients of interest were adults diagnosed with an advanced stage (unresectable or metastatic) HCC who had never received any systemic therapy. Patients were defined as having advanced disease if they had disease progression after or were not eligible for surgical treatments and LRTs. Although a range of ICIs were included in the initial search, only those that were used as first-line treatment were included considering the objective of this study. Studies that evaluated the clinical outcomes, such as median PFS, median OS, objective response rate (ORR), disease control rate (DCR), and treatment-related adverse events (TRAEs), were analyzed. Reviews, case reports, editorials, commentaries, in vitro studies, animal studies, and protocols were excluded.

### 2.2. Data Sources and Search Strategies

We searched three databases, including PubMed, Web of Science, and Embase for eligible studies published from inception to 25 April 2022. The main search terms consisted of “hepatocellular carcinoma”, “immune checkpoint inhibitor”, “PD-1”, “PD-L1”, and “CTLA-4”. Details regarding search strategies were summarized in Appendix A. Conference abstracts and additional references of included studies were also screened.

### 2.3. Study Selection

Two rounds of assessments of search results were conducted independently by two authors. Firstly, titles and abstracts were screened to exclude the irrelevant results. Secondly, potential full texts were reviewed to include eligible studies. All disagreements were discussed between the two authors until they reached a consensus.

### 2.4. Data Extraction

Two authors extracted the following data from each recruited study independently: first author, publication year, study type, characteristics of study patients, region, data source, follow-up time, ICIs agent, comparator, and main clinical outcomes. All disagreements were resolved by discussion between the two authors.

### 2.5. Quality Assessment

Two authors assessed study quality independently using the Newcastle–Ottawa scale (NOS) [27]. Each included study was evaluated according to the main components of NOS, including selection, comparability, and outcome. Total scores ranged from 0 to 9, with higher scores indicating higher quality. Different opinions were discussed between the two authors until they reached a consensus.

### 2.6. Statistical Analysis

Meta-analyses were performed to calculate outcomes of PFS, OS, best responses, and TRAEs. *I*-square and the related *p* value were used to evaluate the heterogeneity. If *I* [2] > 50% or *p* < 0.1, there was a significant heterogeneity [28]. A random effects model was employed for the meta-analysis in these cases, otherwise a common effect model was applied. Stratified analyses were conducted according to different ICIs-based therapy, RECIST criteria, and study characteristics. The outcome for each subgroup was compared with the overall outcome by a one-sample *t* test. Additionally, in order to identify the optimal strategy with superior statistical performance, we further compared subgroups under each type of classification through an independent samples *t* test. The subgroup with the best outcome within each type of classification was selected as the reference group in the comparisons. The relationships between ORR, DCR, PFS, and OS were evaluated by Pearson correlation analysis. Publication bias was assessed by Funnel plot and Egger’s test. Sources of heterogeneity in the estimates were determined by meta-regression analysis. Statistical analysis was accomplished using R studio version 4.1.2 and SPSS version 26.0. [29]. Sensitivity analysis of this study was conducted by omitting each study and calculating the result individually to test the consistency of the pooled results.

## 3. Results

### 3.1. Search Results

Initially, 7805 studies were identified through database searching, of which 2719 were excluded for duplication and 4583 were filtered for study type and language out of scope after title and abstract screening. Then, the full texts of 503 publications were screened thoroughly. A total of 114 studies, which obtained data from research settings that had many strict restrictions, such as randomized controlled trials (RCTs) rather than real-world medical institutions, families, and communities, were excluded. Finally, a total of 38 observational studies were included in the present research (Figure 1).

### 3.2. Quality Assessment

The quality of all 38 studies was assessed using the NOS from three perspectives of selection, comparability, and outcome. Details regarding the assessment were provided in Appendix A. Thirteen studies were evaluated as scoring 9, since they fully met the corresponding scale indicators. The other 25 studies lacked the comparability of cohorts, and thus had lower scores.

### 3.3. Study Characteristics

Appendix A presented the principal characteristics and overall quality of recruited real-world studies. They were mainly conducted in Asia (*n* = 29), with mainland China (*n* = 19) being the most represented region and followed by Taiwan (*n* = 4), Japan (*n* = 3), Republic of Korea (*n* = 2), and Hong Kong (*n* = 1). In addition, USA (*n* = 7), Germany, and Austria (*n* = 1) were the western countries that were involved. Only one study was conducted on a worldwide basis.

#### 3.3.1. Study Designs

A retrospective design was adopted by most studies (*n* = 35), and only three were prospective in design. Data sources were mostly from traditional medical records and EMR (electronic charts, clinical practice database, or otherwise). Specifically, studies were carried out in tertiary care settings, such as hospitals and academic centers. Comparators in most studies were of baseline (pre-drug) measures. Three studies included direct comparisons of ICIs-based combination therapies with tyrosine kinase inhibitors (TKIs), two included direct comparisons of the combination therapies with LRT (TACE) or TKI plus TACE, and four included direct comparisons between different ICIs-based therapies.

#### 3.3.2. Study Populations

In total, 2750 patients with advanced HCC were recruited for the present research. The sample size of included cohorts ranged from 6 to 202. The mean or median age of patients varied from 49.1 to 73 years, and the percentage of male gender varied from 52.8% to 100%. In most studies, HCC patients with HBV or HCV infection were evaluated. Five studies also included NAFLD-related HCC patients, and four included alcoholic etiology. Most patients had Child–Pugh class A liver function and staged at Barcelona Clinic Liver Cancer (BCLC) C. Regarding the ethnicity, most studies (*n* = 29) mainly included Asian, and the remaining studies (*n* = 9) recruited Caucasian, African, and Asian.

#### 3.3.3. Study Drugs and Outcomes Evaluated

ICIs in the setting of first-line treatment involved in this research were anti-PD-(L)1 antibodies. Nivolumab (*n* = 17) was the most common used ICI across studies, followed by camrelizumab (*n* = 9), atezolizumab (*n* = 8), pembrolizumab (*n* = 8), sintilimab (*n* = 7), toripalimab (*n* = 6), tislelizumab (*n* = 2), durvalumab (*n* = 2), cemiplimab (*n* = 1), and AK105 (*n* = 1). ICIs combined with TKIs (*n* = 10), angiogenesis inhibitory monoclonal antibodies (AI mAbs, *n* = 6), LRTs (*n* = 6), and TKIs plus LRTs (*n* = 8) were evaluated. Best response was the most evaluated outcome (*n* = 35), followed by survival outcomes (*n* = 31), TRAEs (*n* = 29), and prognostic factors for PFS and OS (*n* = 14).

### 3.4. Survival Outcomes of ICIs

#### 3.4.1. PFS of ICIs

The final results of 26 real-life studies reported the median PFS of patients receiving first-line monotherapy or combination therapy of ICIs. Three studies reported that the median PFS for first-line ICIs-based therapy (10.2–12.1 months) was better than that for TKIs therapy (5.1–8.9 months) [30,31,32], of which two studies showed statistical differences at *p* < 0.05 [30,32]. One study showed that ICIs-based therapy had significantly better PFS rates of 6 months (93.3%), 12 months (93.3%), and 24 months (77.8%) compared to LRTs therapy (37.5%, 16.7%, and 2.1%, *p* < 0.001) [33]. One revealed that triple therapy (ICIs + TKIs + LRTs) contributed to prolonged PFS (9.2 months) compared with TKI-LRT duotherapy (5.5 months, *p* = 0.006) [34].

A meta-analysis was performed to compare first-line ICIs-based therapy and TKIs therapy in terms of median PFS. We confirmed that ICIs-based therapy (11.34 months; 95% CI: 8.93–13.75) showed PFS benefit compared with TKIs therapy (6.65 months; 95% CI: 4.18–9.11; *p* < 0.01) in real-world practice (Figure 2).

Next, the single-arm and statistical analyses of median PFS included 27 cohorts. The median value of the reported PFS was 6.8 months, which ranged from 2 to 16 months. The pooled median PFS of first-line monotherapy or combination therapy of ICIs was 7.03 (95% CI: 5.55–8.51) months. Subgroup analyses were conducted according to different immune checkpoint targets, ICI drugs, combination modes, and study characteristics. As shown in Table 1, anti-PD-1-based immunotherapy had a better PFS (7.20 (95% CI: 5.12–9.27) months, vs. overall PFS, *p* > 0.05) compared with anti-PD-L1-based immunotherapy (6.66 (95% CI: 5.45–7.87) months, vs. overall PFS, *p* = 0.024). Most anti-PD-1 antibodies involved in this research showed comparable PFS (*p* > 0.05). Meanwhile, there was no statistical difference in PFS outcomes between different ICIs-based combination strategies (*p* > 0.05). Regarding patient characteristics, responders had significantly better median PFS than non-responders (*p* = 0.000), while the same results were observed in patients with or without hepatitis virus infection (*p* > 0.05).

#### 3.4.2. OS of ICIs

The median OS of first-line ICIs-based therapy was reported by 23 studies. One study reported that the median OS of ICIs + TKIs therapy and TKIs therapy were comparable (14.1 months vs. 9.6 months, *p* = 0.105) [32]. However, the triple therapy of ICIs + TKIs + LRTs was proven to be more beneficial to OS when compared with TKIs therapy (Not reached vs. 11.0 months, *p* < 0.001) [30]. The median OS of ICIs + TKIs + LRTs therapy was also better than that of TKIs + LRTs therapy (18.1 months vs. 14.1 months, *p* = 0.004) [34]. Moreover, one study revealed better OS rates of 6 months (93.8%), 12 months (93.8%), and 24 months (80.4%) of ICIs + LRTs therapy compared with LRTs therapy (54.2%, 31.3%, and 8.3%, *p* < 0.001) [33]. Collectively, real-world practice demonstrated that ICIs-based therapy in the first-line setting had better OS benefit in comparison to classic therapy.

In the single-arm analyses, the median value of the reported OS was 13.05 months varying from 3 to 24.8 months, and the pooled median OS for first-line ICIs-based monotherapy or combination therapy was 14.39 (95% CI: 10.91–17.86) months. Further stratified analyses found that, firstly, anti-PD-1-based immunotherapy exhibited no difference in OS outcomes with anti-PD-L1-based immunotherapy (14.54 (95% CI: 10.31–18.78) months vs. 14.9 (95% CI: 13.6–16.3) months, *p* > 0.05), indicating that the majority of ICI drugs presented great effectiveness in real-world practice (Table 2). It was worth noting that with a pooled median OS of 21.22 (95% CI: 16.26–26.17) months, ICIs + TKIs + LRTs therapy had a markedly better OS than monotherapy with ICIs (9.81 (95% CI: 2.18–17.45) months, *p* = 0.026), even if there was no statistical difference between combination therapy and monotherapy (15.98 (95% CI: 12.63–19.33) months vs. 9.81 (95% CI: 2.18–17.45) months, *p* > 0.05). Concerning the treatment response, patients who had a response to ICIs-based therapy showed longer median OS than that of non-responders (15.85 months vs. 2 months, *p* = 0.048).

### 3.5. Prognostic Factors for PFS and OS

Fourteen studies reported prognostic factors for PFS and/or OS of ICIs-based therapy in the setting of first line, and details were summarized in Appendix A. AFP level, ECOG performance status, Child–Pugh grade, etc., were the prognostic factors for PFS in real-world practice. Concerning the OS, the prognostic factors, including gender, age, ECOG performance status, etc., were reported.

### 3.6. Best Response of ICIs

Thirty-five studies were included in the analyses of the best response of first-line ICIs-based therapy in the real world. To determine the response rates of ICIs-based therapy versus TKIs therapy in the first-line setting, meta-analysis was conducted in three studies. ICIs-based therapy showed ORR benefit compared with TKIs therapy, while the DCR was similar in both groups (Figure 3).

The analyses were further stratified by different ICIs-based therapy and RECIST criteria, which were presented in Figure 4, Figure 5, Figure 6 and Figure 7. The ORR and DCR of ICI-based therapy were 0.432 (95% CI: 0.327–0.538) and 0.756 (95% CI: 0.677–0.836), according to mRECIST 1.1, 0.317 (95% CI: 0.218–0.416) and 0.740 (95% CI: 0.644–0.835), judged by RECIST 1.1. The response rates were similar in anti-PD-1-based immunotherapy and anti-PD-L1-based immunotherapy, regardless of whether they were judged by mRECIST 1.1 or RECIST 1.1 (*p* > 0.05, Table 3). However, it was important that the ORR and DCR of ICIs-based combination therapy were better than that of the ICIs monotherapy (ORR: combination therapy vs. overall, *p* > 0.05, monotherapy vs. overall, *p* = 0.01; DCR: combination therapy vs. monotherapy, *p* = 0.002, judged by RECIST 1.1). The best response rate was observed in the ICIs + TKIs + LRTs therapy, since the triple therapy was the only regimen that had a significantly higher response rate than the overall rate (ORR: 0.639 (95% CI: 0.479–0.800) vs. 0.432 (95% CI: 0.327–0.538), *p* = 0.043; DCR: 0.872 (95% CI: 0.800–0.944) vs. 0.756 (95% CI: 0.677–0.836), *p* = 0.016, judged by mRECIST 1.1).

### 3.7. TRAEs of ICIs

Twenty-nine real-world studies were concerned with TRAEs of first-line ICIs-based therapy. The pooled rate of any grade of TRAEs was 0.758 (95% CI: 0.667–0.848) (Figure 8A), grade 1–2 TRAEs was 0.533 (95% CI: 0.455–0.611) (Figure 8B), grade 3–4 TRAEs was 0.188 (95% CI: 0.119–0.258) (Figure 8C), and toxic deaths was 0.002 (95% CI: 0.000–0.010) (Figure 8D). Then, the results of TRAEs were further assessed according to different ICIs-based therapy. The mean rate of any grade of TRAEs was 0.604 (95% CI: 0.515–0.693) in ICIs monotherapy (Figure 8E) and 0.810 (95% CI: 0.722–0.897) in ICIs-based combination therapy (Figure 8F); the mean rate of grade 3–4 TRAEs was 0.143 (95% CI: 0.044–0.242) in ICIs monotherapy (Figure 8G), and 0.202 (95% CI: 0.120–0.284) in ICIs-based combination therapy (Figure 8H). The lower rates of any grade of TRAEs or grade 3–4 TRAEs were observed in ICIs monotherapy compared with ICIs-based combination therapy. In addition, the frequently reported TRAEs were summarized in detail in Appendix A.

### 3.8. Correlations between ORR, DCR, PFS, and OS

The reported ORR, DCR, PFS, and OS for advanced HCC patients receiving first-line ICIs-based therapy were extracted for correlation analyses. As shown in Figure 9A,B, the ORR (R = 0.43, *p* = 0.34) and DCR (R = 0.57, *p* = 0.18) judged by RECIST 1.1 had no correlation with OS. Interestingly, the correlations between ORR, DCR judged by mRECIST 1.1, and OS were 0.85 (*p* = 0.0037) and 0.8 (*p* = 0.01), respectively (Figure 9C,D).

Additionally, the overall correlation was 0.75 (*p* = 0.0019) for PFS and OS (Figure 9E). After conducting subgroup analyses (Figure 9F,G), the PFS was not correlated with OS in ICIs monotherapy (R = 0.87, *p* = 0.056), while it was strongly correlated with OS in ICIs-based combination therapy (R = 0.79, *p* = 0.011).

### 3.9. Assessment of Publication Bias

As shown in Appendix A, a publication bias was observed by Funnel plot and Egger’s test in the analysis of PFS, but not in the analysis of OS. For best response, Funnel plots and Egger’s tests proved significant publication bias in the pooled analysis of DCR judged by mRECIST 1.1 while showing no obvious publication bias in the pooled analyses of ORR judged by mRECIST 1.1 and RECIST 1.1 or DCR judged by RECIST 1.1 (Appendix A). We also used trim and filling method to correct the results of PFS and DCR judged by mRECIST 1.1, and the corrected results were shown in Appendix A.

### 3.10. Sensitivity Analysis

To assess the stability and reliability of the pooled results, sensitivity analysis was conducted and the results were shown in Appendix A. It indicated that no individual study sufficiently influenced the pooled ORR and DCR and only one cohort had an effect on the pooled PFS and OS.

### 3.11. Sources of Heterogeneity

Most meta-analyses were of the high heterogeneity in this research. Meta-regression analysis was used to test for 11 potential sources of heterogeneity in the estimates, including publication year, design of multicenter or single center, viral etiology, age, proportion of male, proportion of patients with Child–Pugh A and BCLC C, sample size, comparator, follow-up time, and quality score. The proportion of patients with Child–Pugh A and BCLC C, as well as the follow-up time, were found to be the significant sources of heterogeneity (*p* < 0.05, Appendix A).

## 4. Discussion

This systematic review and meta-analysis included 38 retrospective and prospective studies to evaluate the effectiveness and safety of first-line ICIs-based therapy for advanced HCC in real-world practice. It revealed diversified regimens involving ICIs in terms of survival outcomes, best response, and TRAEs in real-world practice.

Immune escape is one of the cancer characteristics where PD-1/PD-L1 signaling pathway plays a critical role. The overexpression of PD-L1 on HCC cells results in an increase in interaction of PD-L1 and PD-1 expressed on T cells in the microenvironment of the tumor, which leads to apoptosis and immune anergy. The inhibition of this interaction can enhance immunity reaction against tumor cells [35]. Moreover, CTLA-4 expressed on regulatory T cells modulates the early immunity response. Interfering with the binding between CTLA-4 and B7 in lymph nodes can lead to the increase in activated CD8^+^ cells [35]. Based on these, checkpoint inhibitors have been developed and determined to integrate into clinical practice rapidly [36].

The updated guidelines recommended regimens of atezolizumab plus bevacizumab, sintilimab plus a bevacizumab biosimilar (IBI305), apatinib plus camrelizumab, durvalumab plus tremelimumab, durvalumab, pembrolizumab, nivolumab, lenvatinib plus pembrolizumab or nivolumab, and FOLFOX plus camrelizumab as first-line systemic therapies for advanced HCC, since these immunotherapies have been demonstrated to improve quality of life and prolong the survival in several RCTs [37]. However, this systematic review indicated that, compared with HCC patient samples in RCTs, there were more heterogeneous patients in daily clinical practice (Appendix A). Therefore, the findings about outcomes of ICIs for HCC in real-world practice contributed by supplementing the evidence obtained from RCTs and providing more information about safety signals and clinical outcomes. The evidence from real-world practice could become a critical component of the overall evidence for clinical practice [21]. However, most existing real-world studies regarding first-line ICIs-based therapy for advanced HCC were retrospective, single center, based on a small sample, and noncomparative, as well as having different endpoints in design, resulting in a high risk of bias. Consequently, future prospective, multicentric, comparative, standardized endpoints and high-quality studies are required.

In this research, the scope of use for first-line ICIs-based therapy in real-world practice was found to expand beyond recent clinical guidelines, which provided more evidence for treatment strategies for advanced HCC. The pooled results of this research demonstrated that ICIs-based therapy in the first-line setting witnessed better outcomes in survival and response rate from real-world practice compared with TKIs therapy, which was also recommended as a first-line regimen by clinical guidelines. Furthermore, the combination therapy of ICIs, TKIs, plus LRTs deserved the optimal ICIs-based regimen in real-world practice, showing the best outcomes of OS, ORR, and DCR. Similarly, in our previous study that evaluated the efficacy of first-line ICIs-based therapy for unresectable HCC patients in clinical practice, we found longer OS in ICIs-based therapy compared with sorafenib [37]. Although the study showed that the combination mode of ICIs + AI mAbs had the best response rate in terms of clinically, the combination therapy would be a new trend regardless in clinical or in real-world practice [37]. Moreover, it was reported that less than 20% of advanced HCC patients steadily responded to the monotherapy of ICIs [19,20]. The combination therapy was indeed a new treatment option, for instance, anti-vascular endothelial growth factor (VEGF) could reverse immunosuppression mediated by VEGF and promote tumor lysis mediated by T cells to enhance the efficiency of ICIs [38,39]. Chemoembolization has previously been demonstrated to induce the spread of tumor-associated antigens and increase VEGF, which provided a strong reason for combination therapy with immunostimulating agents. In this regard, the triple therapy of ICIs, chemoembolization, plus molecular target agents possessing anti-VEGF effect was proposed [36]. Thus, ICIs-based combination therapy, especially triple therapy of ICIs, TKIs, plus LRTs, could be the direction of clinical exploration in the future, owing to the relatively limited effectiveness of monotherapy.

Several studies evaluated the relationships between OS and early endpoints, such as ORR, DCR, and PFS, in cancer patients treated with ICIs [40,41,42]. The results found a poor correlation between response rate and OS, and the correlation coefficient of 6-month PFS and 12-month OS ranged from 0.74 to 0.89 [40,41,42]. When analyzing the associations of response rate and OS, our results based on RECIST 1.1 were consistent with previous conclusions. However, the calculated overall correlation coefficients of ORR and DCR judged by mRECIST 1.1 with OS were relatively high, indicating that the associations of these endpoints were stronger. These findings suggested that first-line ICIs-based therapy for advanced HCC from real-world practice had a better effectiveness in disease control assessed using mRECIST 1.1, which might correlate to prolonged survival. Concerning correlation analyses of PFS and OS, our study showed that the correlation of PFS and OS in ICIs-based combination therapy was better than that in ICIs monotherapy. Responders’ proportion in ICIs-based combination therapy was greatly higher than those in monotherapy of ICIs. The analyses suggested that advanced HCC patients in the real world who had no progressive disease might obtain improved survival from first-line combination therapy. Therefore, ORR, DCR judged by mRECIST 1.1, and PFS could serve as potential prognostic factors for OS in advanced HCC patients who received first-line ICIs-based therapy in a real-world practice.

Taken together, future studies focusing on ICIs-based regimens as first-line systemic therapy for advanced HCC are suggested to adopt prospective and multicentric design and set control group for comparison, such as classic first-line therapy or other first-line ICIs-based therapy. Regarding study regimens, ICIs-based combination therapies, particularly triple therapy of ICIs, TKIs, plus LRTs, are recommended. ORR, DCR judged by mRECIST 1.1, and PFS are determined as early endpoints, since they are accepted as potential surrogate markers for OS.

Concerning the quality of included studies, since most of the study designs were a single group assignment, the scores of quality assessment were varied. However, all the studies were included in the final analyses for the reason that we aimed to consider more ICIs-based treatment strategies in real-world practice. The sensitivity analysis demonstrated that the stability of the pooled results of PFS and OS were influenced by the poor quality of some studies. Therefore, future real-world studies should attach importance to the comparability of cohorts to improve the study quality.

Moreover, we identified the significant sources of heterogeneity in this research. The proportion of patients with Child–Pugh A and BCLC C, as well as the follow-up time, could contribute to the heterogeneities in pooled analyses of DCR, ORR judged by RECIST 1.1, and median OS, respectively. It, consequently, suggests that the real-world studies with large sample sizes are needed to further test the impacts of these sources of heterogeneity on the related outcomes by conducting appropriate subgroup analysis.

To our knowledge, this research provided the first systematic analysis of clinical outcomes associated with ICIs-based therapy for advanced HCC in the first-line setting from real-world practice. However, several limitations still existed. Firstly, some of the recruited studies were conference abstracts, resulting in limited data of baseline and analyses of subgroups. Secondly, the majority of the included studies were retrospective studies, which presented certain levels of heterogeneity and publication bias. Thirdly, the characteristics of patients varied among different studies, which might lead to ineluctable bias for analysis. Finally, there is still a shortage of direct outcome comparisons of different ICIs-based therapies, which requires more RCTs and real-world studies in the future.

## 5. Conclusions

This systematic literature review and meta-analysis revealed diversified first-line ICIs-based therapies for advanced HCC in real-world practice. Future studies are needed to adopt prospective, multicentric, and comparative designs to test the ICIs-based combination therapies, especially the triple therapies of ICIs, TKIs, plus LRTs, which could contribute to better survival outcomes and response rates compared with monotherapy.

## Figures and Tables

**Figure 1 cancers-15-00260-f001:**
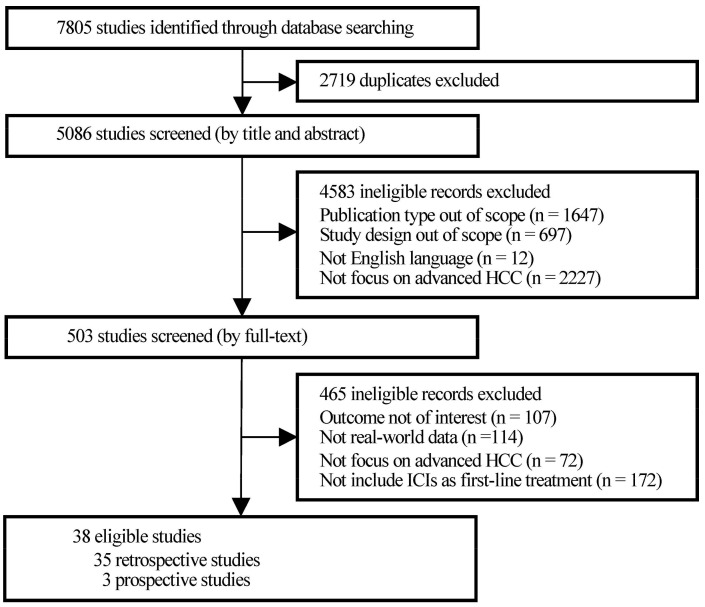
PRISMA flowchart of study selection.

**Figure 2 cancers-15-00260-f002:**
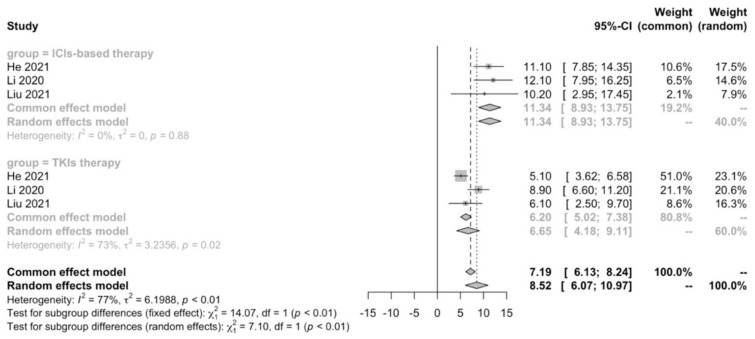
Meta-analysis of PFS in advanced HCC patients receiving first-line ICIs-based therapy versus TKIs therapy.

**Figure 3 cancers-15-00260-f003:**
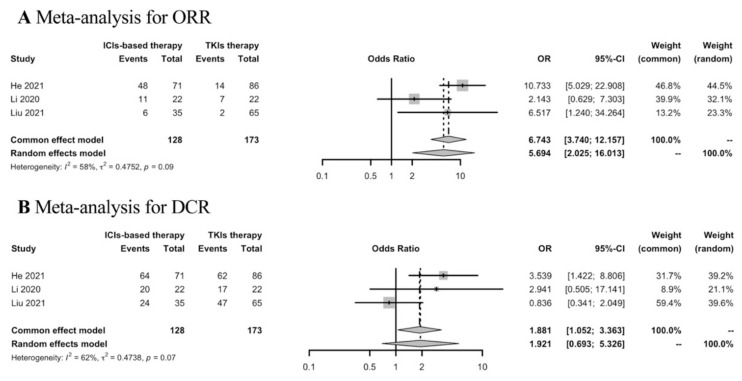
Meta-analysis of response rates in advanced HCC patients receiving first-line ICIs-based therapy versus TKIs therapy.

**Figure 4 cancers-15-00260-f004:**
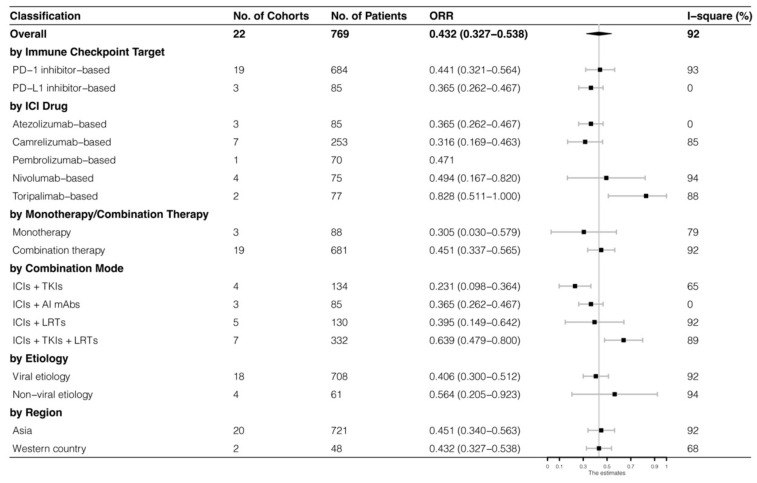
Subgroup analysis of ORR judged by mRECIST 1.1.

**Figure 5 cancers-15-00260-f005:**
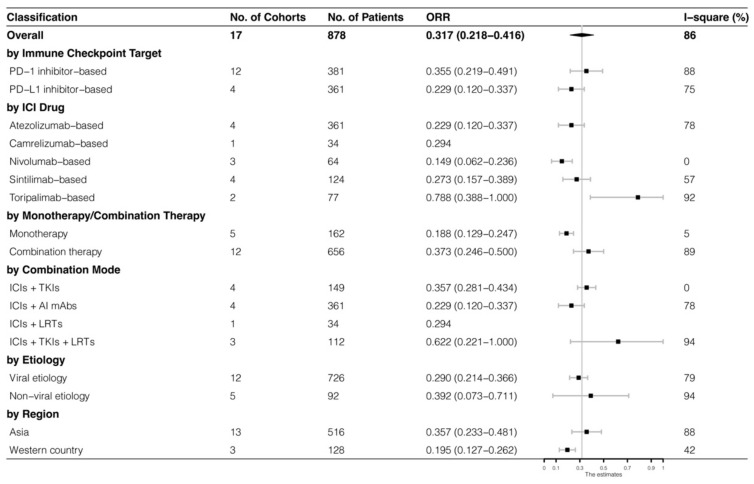
Subgroup analysis of ORR judged by RECIST 1.1.

**Figure 6 cancers-15-00260-f006:**
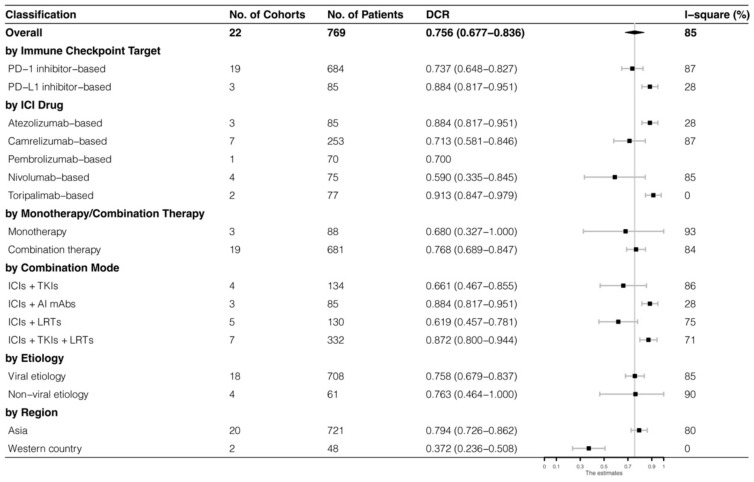
Subgroup analysis of DCR judged by mRECIST 1.1.

**Figure 7 cancers-15-00260-f007:**
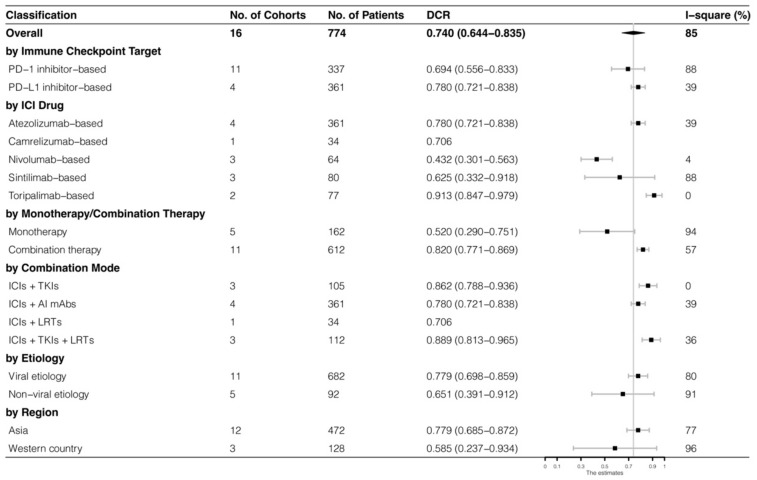
Subgroup analysis of DCR judged by RECIST 1.1.

**Figure 8 cancers-15-00260-f008:**
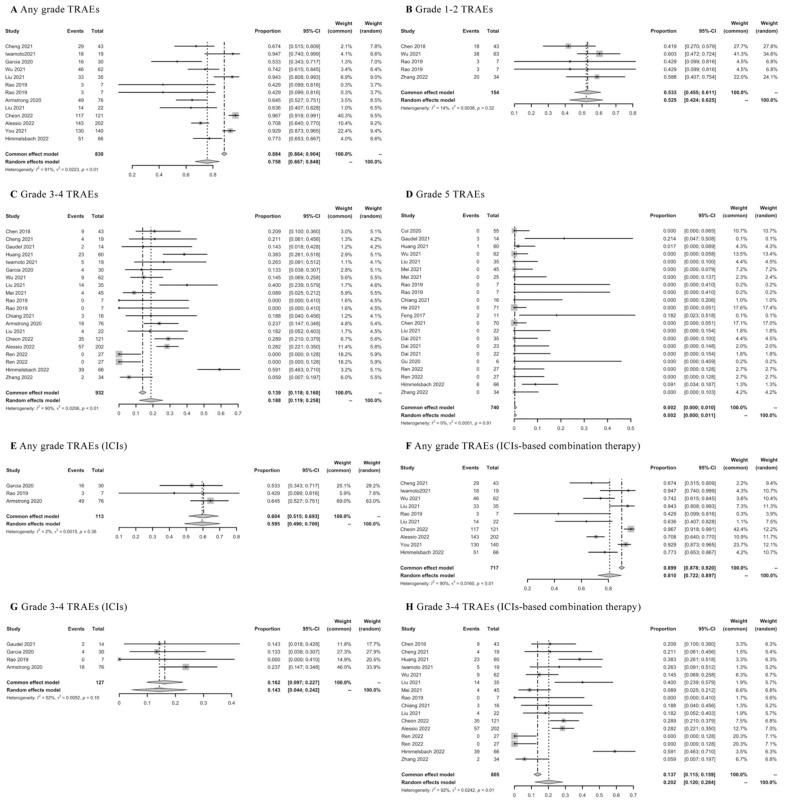
Meta-analysis of TRAEs in advanced HCC patients receiving first-line ICIs-based therapy.

**Figure 9 cancers-15-00260-f009:**
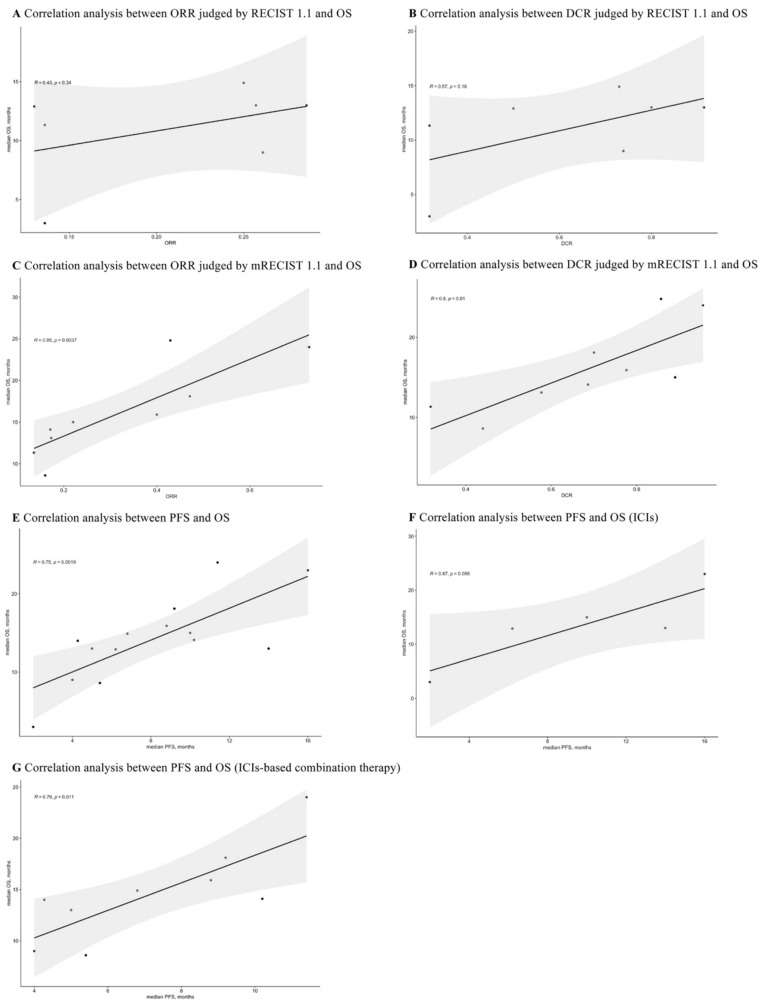
Correlation analyses between ORR, DCR, PFS, and OS.

**Table 1 cancers-15-00260-t001:** Subgroup analysis of PFS in advanced HCC patients receiving first-line ICIs-based therapy.

Classification	No. of Cohorts	No. of Patients	Median Value of PFS, Months	Ranges of PFS, Months	Median PFS (95% CI), Months	*I* [2], %	*p* Value	
							Overall	Subgroup
Overall	27	1509	6.8	2–16	7.03 (5.55–8.51)	94	RG	/
by Immune Checkpoint Target								
PD-1 inhibitor-based	22	1036	7.25	2–16	7.20 (5.12–9.27)	93	0.946	RG
PD-L1 inhibitor-based	4	408	6.5	5.4–6.8	6.66 (5.45–7.87)	0	0.024	0.807
by ICI Drug								
Atezolizumab-based	4	408	6.5	5.4–6.8	6.66 (5.45–7.87)	0	0.024	0.006
Camrelizumab-based	6	296	8.05	3–11.4	8.19 (3.64–12.75)	93	0.937	0.385
Nivolumab-based	3	111	6.2	4–6	8.73	/	0.79	0.779
Pembrolizumab-based	1	70	9.2	/	9.2 (7.1–10.4)	/	/	/
Sintilimab-based	4	124	4.5	2–8.6	3.46 (1.69–5.22)	88	0.144	0.138
Toripalimab-based	1	71	11.1	/	11.1 (7.85–14.35)	/	/	RG
by Monotherapy/Combination Therapy								
Monotherapy	6	253	8.1	2–16	8.02 (0.99–15.05)	92	0.654	RG
Combination therapy	21	1256	6.8	3–12.1	6.99 (5.55–8.43)	86	0.596	0.576
by Combination Mode								
ICIs + TKIs	8	366	7.25	4–12.1	6.86 (4.32–9.40)	78	0.836	0.301
ICIs + AI mAbs	4	408	6.5	5.4–6.8	6.66 (5.45–7.87)	0	0.024	0.073
ICIs + LRTs	4	239	5.19	3–10	4.14 (2.06–6.22)	64	0.329	0.124
ICIs + TKIs + LRTs	5	243	9.2	5–11.4	9.06 (6.15–11.97)	84	0.264	RG
by Etiology								
Viral etiology	24	1391	6.9	2–16	6.81 (5.26–8.36)	94	0.91	0.734
Non-viral etiology	3	118	6.5	6.2–12.1	7.03 (5.55–8.51)	80	0.766	RG
by Treatment Response								
Responder	2	/	/	10.5–11	10.75	/	0.051	RG
Non-responder	3	/	2	1–2.3	1.77	/	0.004	0.000
by Region								
Asia	21	1065	7	2–12.1	6.90 (5.23–8.57)	94	0.569	0.234
Western country	5	242	6.5	4–16	7.08 (5.50–8.66)	73	0.507	RG

Abbreviations: AI mAbs, angiogenesis inhibitory monoclonal antibodies; ICI, immune checkpoint inhibitor; LRT, locoregional therapy; PD-1, programmed cell death-1; PD-L1, programmed cell death ligand-1; PFS, progression-free survival; RG, reference group; TKI, tyrosine kinase inhibitor; /, not available. *p* value: the outcome for each subgroup was compared with the overall outcome, and subgroups were compared with the reference group within each type of classification.

**Table 2 cancers-15-00260-t002:** Subgroup analysis of OS in advanced HCC patients receiving first-line ICIs-based therapy.

Classification	No. of Cohorts	No. of Patients	Median Value of OS, Months	Ranges of OS, Months	Median OS (95% CI), Months	*I* [2], %	*p* Value	
							Overall	Subgroup
Overall	24	1246	13.05	3–24.8	14.39 (10.91–17.86)	97	RG	/
by Immune Checkpoint Target								
PD-1 inhibitor-based	20	811	13.05	3–24.8	14.54 (10.31–18.78)	97	0.862	0.78
PD-L1 inhibitor-based	1	202	14.9	/	14.9 (13.6–16.3)	/	/	RG
by ICI Drug								
Atezolizumab-based	1	202	14.9	/	14.9 (13.6–16.3)	/	/	0.717
Camrelizumab-based	6	350	14.05	13.1–24.8	18.91 (12.59–25.23)	69	0.156	RG
Nivolumab-based	6	147	10.17	5–23	11.54	/	0.495	0.128
Pembrolizumab-based	1	70	18.1	/	18.1 (16.5–20.7)	/	/	0.889
Sintilimab-based	3	80	9	3–13	5.87 (0–11.74)	94	0.223	0.054
by Monotherapy/Combination Therapy								
Monotherapy	9	332	11.33	3–23	9.81 (2.18–17.45)	92	0.273	0.107 0.026 (vs. ICIs + TKIs + LRTs)
Combination therapy	15	914	14	8.6–24.8	15.98 (12.63–19.33)	85	0.282	RG
by Combination Mode								
ICIs + TKIs	6	292	12.7	8.6–18.1	9.20 (1.51–16.88)	86	0.566	0.032
ICIs + AI mAbs	1	202	14.9	/	14.9 (13.6–16.3)	/	/	0.49
ICIs + LRTs	3	191	13.3	9–14	12.1	/	0.489	0.074
ICIs + TKIs + LRTs	5	228	18.1	13–24.8	21.22 (16.26–26.17)	75	0.066	RG
by Etiology								
Viral etiology	20	1180	13.66	3–24.8	14.39 (10.91–17.86)	97	0.541	RG
Non-viral etiology	4	66	10.17	5–12.9	9.56	/	0.11	0.125
by Treatment Response								
Responder	2	/	/	12.7–19	15.85	/	0.581	RG
Non-responder	2	/	/	2–2	2	/	/	0.048
by Region								
Asia	16	821	13.65	3–24.8	14.54 (10.76–18.33)	97	0.654	RG
Western country	7	223	11.33	5–23	13 (7.9–18.1)	/	0.471	0.376

Abbreviations: AI mAbs, angiogenesis inhibitory monoclonal antibodies; ICI, immune checkpoint inhibitor; LRT, locoregional therapy; PD-1, programmed cell death-1; PD-L1, programmed cell death ligand-1; PFS, progression-free survival; RG, reference group; TKI, tyrosine kinase inhibitor; /, not available. *p* value: the outcome for each subgroup was compared with the overall outcome, and subgroups were compared with the reference group within each type of classification.

**Table 3 cancers-15-00260-t003:** Statistical analysis of response rates in advanced HCC patients receiving first-line ICIs-based therapy.

	ORR					
	mRECIST 1.1			RECIST 1.1		
Classification	Rate (95% CI)	*p* Value		Rate (95% CI)	*p* Value	
		Overall	Subgroup		Overall	Subgroup
Overall	0.432 (0.327–0.538)	RG	/	0.317 (0.218–0.416)	RG	/
by Immune Checkpoint Target						
PD-1 inhibitor-based	0.441 (0.321–0.564)	0.779	RG	0.355 (0.219–0.491)	0.616	RG
PD-L1 inhibitor-based	0.365 (0.262–0.467)	0.138	0.43	0.229 (0.120–0.337)	0.088	0.262
by ICI Drug						
Atezolizumab-based	0.365 (0.262–0.467)	0.138	0.17	0.229 (0.120–0.337)	0.088	0.199
Camrelizumab-based	0.316 (0.169–0.463)	0.223	0.018	0.294	/	0.39
Pembrolizumab-based	0.471	/	0.416	/	/	/
Nivolumab-based	0.494 (0.167–0.820)	0.698	0.265	0.149 (0.062–0.236)	0.067	0.191
Sintilimab-based	/	/	/	0.273 (0.157–0.389)	0.516	0.026
Toripalimab-based	0.828 (0.511–1.000)	0.24	RG	0.788 (0.388–1.000)	0.256	RG
by Monotherapy/Combination Therapy						
Monotherapy	0.305 (0.030–0.579)	0.59	0.487	0.188 (0.129–0.247)	0.01	0.102 0.029 (vs. ICIs + TKIs + LRTs)
Combination therapy	0.451 (0.337–0.565)	0.795	RG	0.373 (0.246–0.500)	0.429	RG
by Combination Mode						
ICIs + TKIs	0.231 (0.098–0.364)	0.121	0.014	0.357 (0.281–0.434)	0.331	0.208
ICIs + AI mAbs	0.365 (0.262–0.467)	0.138	0.043	0.229 (0.120–0.337)	0.088	0.068
ICIs + LRTs	0.395 (0.149–0.642)	0.806	0.117	0.294	/	0.506
ICIs + TKIs + LRTs	0.639 (0.479–0.800)	0.043	RG	0.622 (0.221–1.000)	0.272	RG
by Etiology						
Viral etiology	0.406 (0.300–0.512)	0.576	0.238	0.290 (0.214–0.366)	0.402	0.538
Non-viral etiology	0.564 (0.205–0.923)	0.493	RG	0.392 (0.073–0.711)	0.654	RG
by Region						
Asia	0.451 (0.340–0.563)	0.258	RG	0.357 (0.233–0.481)	0.573	RG
Western country	0.432 (0.327–0.538)	0.324	0.286	0.195 (0.127–0.262)	0.076	0.223
	DCR					
	mRECIST 1.1			RECIST 1.1		
Classification	Rate (95% CI)	*p* Value		Rate (95% CI)	*p* Value	
		Overall	Subgroup		Overall	Subgroup
Overall	0.756 (0.677–0.836)	RG	/	0.740 (0.644–0.835)	RG	/
by Immune Checkpoint Target						
PD-1 inhibitor-based	0.737 (0.648–0.827)	0.765	0.392	0.694 (0.556–0.833)	0.616	0.281
PD-L1 inhibitor-based	0.884 (0.817–0.951)	0.278	RG	0.780 (0.721–0.838)	0.129	RG
by ICI Drug						
Atezolizumab-based	0.884 (0.817–0.951)	0.278	0.295	0.780 (0.721–0.838)	0.129	0.019
Camrelizumab-based	0.713 (0.581–0.846)	0.576	0.108	0.706	/	0.215
Pembrolizumab-based	0.700	/	0.21	/	/	/
Nivolumab-based	0.590 (0.335–0.845)	0.333	0.145	0.432 (0.301–0.563)	0.044	0.010
Sintilimab-based	/	/	/	0.625 (0.332–0.918)	0.574	0.194
Toripalimab-based	0.913 (0.847–0.979)	0.149	RG	0.913 (0.847–0.979)	0.135	RG
by Monotherapy/Combination Therapy						
Monotherapy	0.680 (0.327–1.000)	0.743	0.534	0.520 (0.290–0.751)	0.126	0.002 0.041 (vs. ICIs + TKIs + LRTs)
Combination therapy	0.768 (0.689–0.847)	0.795	RG	0.820 (0.771–0.869)	0.006	RG
by Combination Mode						
ICIs + TKIs	0.661 (0.467–0.855)	0.431	0.036	0.862 (0.788–0.936)	0.161	0.388
ICIs + AI mAbs	0.884 (0.817–0.951)	0.278	0.618	0.780 (0.721–0.838)	0.129	0.068
ICIs + LRTs	0.619 (0.457–0.781)	0.207	0.033	0.706	/	0.234
ICIs + TKIs + LRTs	0.872 (0.800–0.944)	0.016	RG	0.889 (0.813–0.965)	0.161	RG
by Etiology						
Viral etiology	0.758 (0.679–0.837)	0.932	0.847	0.779 (0.698–0.859)	0.498	RG
Non-viral etiology	0.763 (0.464–1.000)	0.912	RG	0.651 (0.391–0.912)	0.601	0.473
by Region						
Asia	0.794 (0.726–0.862)	0.316	RG	0.779 (0.685–0.872)	0.529	RG
Western country	0.372 (0.236–0.508)	0.089	0.003	0.585 (0.237–0.934)	0.504	0.211

Abbreviations: AI mAbs, angiogenesis inhibitory monoclonal antibodies; DCR, disease control rate; ICI, immune checkpoint inhibitor; LRT, locoregional therapy; ORR, objective response rate; PD-1, programmed cell death-1; PD-L1, programmed cell death ligand-1; PFS, progression-free survival; RG, reference group; TKI, tyrosine kinase inhibitor; /, not available. *p* value: the outcome for each subgroup was compared with the overall outcome, and subgroups were compared with the reference group within each type of classification.

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
