# Peer review of "Clinical Outcomes Associated with Monotherapy and Combination Therapy of Immune Checkpoint Inhibitors as First-Line Treatment for Advanced Hepatocellular Carcinoma in Real-World Practice: A Systematic Literature Review and Meta-Analysis"

_cancers, 2022, doi:10.3390/cancers15010260_

Round 1

Reviewer 1 Report

General comment : this is a comprehensive review on the potential role of immune check poit inhibitors as first-line treatment of advanced hepatocellular carcinoma (HCC). As such, the results are convincing and as the authors state, diversified (or mitigated) across this population  of patients. The paper is well  written.

Specific comment: the authors do not state why a patient with HCC is advanced and may not be amenable to other available therapy, including: resection or non-surgical resection i.e. radiofrequency/ablation, liver transplantation, TACE or radioembolization. This may be a huge bias with the question raised.

The studied population is mainly of eastern Asian origin, with little information on western studies: no information on these latter studies ia available. If they contain mainly  Asian patients abroad, then the sample is homogeneous. Otherwise, the seven papers from  USA and one from Australia should be omitted, and the paper dedicated solely to Eastern Asia data.

Finally, the authors recognize a significant heterogeneity in their analysis but do not really discuss it, and especially to what extend this may impact on the final conclusion, which remains vague…

Reviewer 2 Report

This systematic review aimed to address an interesting and essential topic regarding the real-world efficacy of immune checkpoint inhibitor therapy in advanced HCC. There is an evident need to understand the outcomes of mono and combinatorial therapies outside of clinical trials.  

However, some of the reported analyses were not appropriate given the eligibility criteria and available patient cohorts. For instance, the subgroup comparison of PD-1 vs PD-L1 was not appropriate as for PD-L1 the only inhibitor included was Atezolizumab which based on the Table S4 was always administrated in combination with Bevacizumab. The same applies to the ICI drug subgroup comparisons. 

Major issues:

Line 127-128: In this analysis, the authors compared the subgroups with the overall outcome instead of the individual subgroups and the rationale of this choice is not justified. In fact, this analysis lacks to inform if there is a difference between subgroups. Moreover, the authors should further explain the subcategorization comparisons and specify if this was done with an independent samples t test for all subgroup combinations E.g. ICI + TKIs vs ICI + AI mAbs or ICI + TKIs vs ICI + LRTs? From Tables 1, 2, 3 it looks like there was one reference group (Toripalimab or Sintilimab for instance) compared to the other subgroups (ICI drugs). 

It would be more informative to report the results on all possible comparisons rather than selecting just one reference group, or at least justify the choice of the RG.

Line 144: Please add the summary of the study quality assessment and specify how this assessment affected your data analysis. 

Line 203-210: Anti-PD-1 and anti-PD-L1 subgroups cannot be compared in this manner. Based on the supplementary information provided, table S4 – characteristics of included studies, all patients who received anti-PD-L1 (atezolizumab) also received Bevacizumab in combination. These patients cannot therefore be considered as part of any monotherapy group, thus making the subgroup comparison between anti-PD1 and anti PD-L1 groups misleading. In addition to this, the subgroup comparisons between ICI drugs (table 1) suggests that these were all administered as monotherapies, but as mentioned before in many cases these were part of a combination regimen. All the above also applies to the analysis of median OS reported in Table 2 (page 9) and response rates reported in Table 3

Discussion: The authors recently published a similar systematic review on mono and combination therapy of immune checkpoint inhibitors as first-line treatment for unresectable HCC (cited in this manuscript, 36). It would be appropriate to further highlight in the discussion the differences between the studies and the findings.

Minor points:

Line 46: Reference 1 is for a paper from 2016 which cites a study published in 2013 for this claim. Needs a more recent reference. 

Line 51: ‘smoke’ correct to ‘smoking’ 

Line 136: Please add the date of the last literature search. 

Line 151: “USA” instead of “America” 

Line 175: ‘Studies with ICIs combined with TKIs, angiogenesis inhibitory monoclonal anti- 175 bodies (AI mAbs), and LRTs were 10, 6, and 6, respectively, while triple therapies of ICIs, 176 TKIs, plus LRTs were evaluated in 8 studies.’ – This can be rewritten in a much simpler sentence e.g. put number of studies in brackets next to therapy type. 

Line 182: Sentence incomplete. 

Page 9: homogenise data in textsome data has month inside brackets, other do not. 

Page 20, line 8: ‘no’ correct to ‘not’

Figure 8: Please increase the size of forest plots.

Reviewer 3 Report

Manuscript ID cancers-2029040, entitled " Clinical outcomes associated with monotherapy and combination therapy of immune checkpoint inhibitors as first-line treatment for advanced hepatocellular carcinoma in real-world practice: a systematic literature review and meta-analysis”.

Thank you for the opportunity to review this manuscript.

The study by Huimin Zou et al deals with topic of interest and a question that, although not new, it is still under investigation and very pertinent to our clinical practice.

The Authors provided the first systematic analysis of clinical outcomes associated with ICIs-based therapy for advanced HCC in the first-line setting from real-world practice.

The study design and the analysis are appropriate.

All authors are from a single discipline? Has there been any involvement of a statistician? Were hepatologists and oncologist involved in the study? 

The manuscript appears to be very long and many aspects in the results section would benefit from being streamlined and shortened (with reference to the tables).

Round 2

Reviewer 1 Report

The authors must define what the meaning is of 'advanced stage' (BCLC or any other system, but not a single criterion, especially in a meta-analysis. Otherwise, quality assessment is not possible. I will be strict on this.

Round 3

Reviewer 1 Report

I am still not convinced with the definition of 'advanced HCC'. A recent report of the BCLC adapted guidelines is clear about this (Annals of Surgical Oncology volume 29pages7231–7234 (2022)), but the authors remain elusive about this.

Even if I am sure that patients with distant metastatic disease are well to be considered, I am not sure all other patients may not benefit from liver transplantation after tumor shrinkage, TACE, SIRT or other options. To be credible, this review should clearly define which patients are not anymore amenable to any other treatments. Maybe ICI may be a cost-effective and efficient option in many of these patients, but this should not be studied in 'real-life' but well in randomized trials.

Therefore I believe this systematic review is hugly biased.

Thanks to the authors for correctly replying to the other comments.